# Laboratory Investigation of Tomography-Controlled Continuous Steel Casting

**DOI:** 10.3390/s22062195

**Published:** 2022-03-11

**Authors:** Ivan Glavinić, Imamul Muttakin, Shereen Abouelazayem, Artem Blishchik, Frank Stefani, Sven Eckert, Manuchehr Soleimani, Iheb Saidani, Jaroslav Hlava, Saša Kenjereš, Thomas Wondrak

**Affiliations:** 1Department of Magnetohydrodynamics, Institute of Fluid Dynamics, Helmholtz-Zentrum Dresden-Rossendorf, Bautzner Landstraße 400, 01328 Dresden, Germany; f.stefani@hzdr.de (F.S.); s.eckert@hzdr.de (S.E.); i.saidani@hzdr.de (I.S.); t.wondrak@hzdr.de (T.W.); 2Engineering Tomography Laboratory, Department of Electronic and Electrical Engineering, University of Bath, Claverton Down, Bath BA2 7AY, UK; i.muttakin@bath.ac.uk (I.M.); ms350@bath.ac.uk (M.S.); 3Faculty of Mechatronics, Informatics and Interdisciplinary Studies, Technical University of Liberec, Studentská 1402/2, 46117 Liberec, Czech Republic; shereen.abouelazayem@tul.cz (S.A.); jaroslav.hlava@tul.cz (J.H.); 4Department of Chemical Engineering, Faculty of Applied Sciences, Delft University of Technology, van der Maasweg 9, 2628 CD Delft, The Netherlands; a.blishchik@tudelft.nl (A.B.); s.kenjeres@tudelft.nl (S.K.)

**Keywords:** contactless inductive flow tomography, continuous casting, clogging, flow control, EMBr, inductive measurements, mini-LIMMCAST

## Abstract

More than 96% of steel in the world is produced via the method of continuous casting. The flow condition in the mould, where the initial solidification occurs, has a significant impact on the quality of steel products. It is important to have timely, and perhaps automated, control of the flow during casting. This work presents a new concept of using contactless inductive flow tomography (CIFT) as a sensor for a novel controller, which alters the strength of an electromagnetic brake (EMBr) of ruler type based on the reconstructed flow structure in the mould. The method was developed for the small-scale Liquid Metal Model for Continuous Casting (mini-LIMMCAST) facility available at the Helmholtz-Zentrum Dresden-Rossendorf. As an example of an undesired flow condition, clogging of the submerged entry nozzle (SEN) was modelled by partly closing one of the side ports of the SEN; in combination with an active EMBr, the jet penetrates deeper into the mould than when the EMBr is switched off. Corresponding flow patterns are detected by extracting the impingement position of the jets at the narrow faces of the mould from the CIFT reconstruction. The controller is designed to detect to undesired flow condition and switch off the EMBr. The temporal resolution of CIFT is 0.5 s.

## 1. Introduction

When it comes to controlling industrial processes, it is paramount to have reliable and accurate measurements and a detailed understanding of the underlying process. However, in the case of continuous casting of steel, due to the aggressive environment, measuring any variable of interest is challenging. In this process, liquid steel is brought by ladles and poured in the tundish, which acts as a buffer storage between ladle changes. From the tundish, steel flows through the Submerged Entry Nozzle (SEN) into the water-cooled mould where the initial solid shell is formed. A solid shell with a liquid core is continuously extruded from the underside of the mould and further cooled and guided to the subsequent processes. Conditions during casting and the flow pattern in the mould are relevant for the quality of the end products [1,2,3]. In order to influence the flow in the mould, different electromagnetic actuators were developed in the last decades, which do not require any direct contact to the melt. Typical systems are electromagnetic stirrers and electromagnetic brakes (EMBrs) [4]. Systems with different magnetic field shapes for slab casters are available: the ruler and double-ruler EMBr, local braking, electromagnetic stirring at the meniscus and/or at the strand, and a combination of a stirrer at the meniscus level and an EMBr of ruler-type below the SEN. The control of these actuators based on the current flow structure in the mould would be desirable.

However, opaqueness and high temperatures of the liquid steel make it nearly impossible to use conventional flow measurement methods. The nail-bed dipping technique is widely used to get a coarse instantaneous subsurface velocity profile [5,6], and measurements of the inclination of an immersed paddle yield continuous but localized information of the velocity [7]. Both methods are difficult to be used for online monitoring. Contactless methods for local velocity measurements exploit the high electrical conductivity of the liquid steel and rely on the principle of induction. Recently, Lorentz force velocimetry was tested to monitor the meniscus velocity in a steel caster [8]. The downside of localized velocity measurements is that information is provided only at a single point or small region in space, while several sensors would be needed to capture the overall flow pattern. There is a need for multidimensional measurements that can provide information-rich data for measurement during casting. Recently, a new measurement technique for temperature distribution in the copper walls of the mould using fiber Bragg gratings [9,10] became available. This technique allows to detect the spatially resolved shape of the meniscus in real time. From the shape of the meniscus, some general assumptions of the flow structure in the mould can be made. A different approach is the measurement of the local velocity and the use of numerical simulations to infer the velocity structure in the mould. Zhang et al. use an immersed paddle to measure the submeniscus flow velocity in connection with a complex mathematical model that combines computational fluid dynamics and discrete phase method [11]. Hashimoto et al. developed a real-time flow estimation algorithm based on three-dimensional transient modelling in order to obtain information on the steel flow. However, this approach was validated by simulations only [12].

In the absence of direct measurements of the flow field, the EMBrs are usually controlled by process parameters or by product recipes. Due to the limited data from the real process, the operation parameters are identified using process modelling techniques, either numerical or physical, which in turn enables the development of new strategies and measurement methods. Numerical methods can provide deep knowledge of physical processes and phenomena, but they require a great amount of computational resources for just a single scenario [13,14]. Thus, it is complicated to use them in order to test new control methodologies. Furthermore, they need to be validated using experimental data provided by the physical models. By contrast, physical models also enable much larger parameter studies [15]. Specifically, for continuous casting, we can distinguish between two types of physical models: water models [16] and liquid metal models [17,18]. Both types have their advantages and disadvantages. Water models are easier to operate and built on a 1:1 scale, and optical methods can be employed for flow measurement. However, since water is electrically nonconductive, it is impossible to model the effects of the electromagnetic actuators. Liquid metal models allow for such investigation, but they are more difficult to build, maintain, and operate.

The development of control loops for EMBr based on real-time flow measurements just started quite recently. Dekemele et al. investigated the feasibility to develop a control loop for an electromagnetic stirrer using a travelling magnetic field in the region of the jet based on submeniscus flow measurements with an immersed paddle [19]. Recently, a successful approach of controlling the strength of an EMBr for a thin caster by measuring the shape of the meniscus using fiber Bragg gratings was reported [20]. Other control variables, such as stopper rod position, amount of argon gas, and even casting speed, could, in theory, be modulated, but their master controller is part of the system ensuring safe and stable operation, i.e., position of the stopper rod is controlled by the mould level controller.

The use of inductive tomographic sensors for continuous casting control has not yet been studied. Lately, two different inductive measurement modalities were developed, mutual inductance tomography (MIT), which can provide spatially resolved distribution of gas/liquid distribution in the SEN [21], and contactless inductive flow tomography (CIFT), which can reconstruct the flow structure of the liquid steel in the mould [22]. While these techniques are not yet used in the production environment, they present interesting opportunities to visualize the global three-dimensional flow in the mould and the SEN in real time.

MIT is able to reconstruct the conductivity distribution in one cross-section of the SEN, and thereby distinguish between liquid metal and argon gas in case of a two-phase flow in a physical model [23,24]. Tomographic data/images as well as SEN filling profiles would be useful data for the control implementation. However, the challenges remain for dynamic liquid metal two-phase flow measurements. Small distributed inner bubbles are difficult to detect and reconstruct. The multifrequency method, bypassing reconstruction and using just raw data, or data-driven approaches for quantification are the future directions that could be investigated for continuous casting [25].

CIFT relies on measuring the perturbations of an applied magnetic field (primary field) caused by the movement of the conductive fluid [22,26]. The potential of CIFT to reconstruct the velocity structure has been demonstrated in the laboratory model of a continuous caster [27], as well as in other laboratory experiments that model Rayleigh-Benard convection and Czochralski crystal growth [28]. The main advantage of CIFT, in comparison to other measurement techniques, is that the measurements of the magnetic field are done outside the fluid, and it is able to reconstruct the essential structures of the global flow field. However, the challenge is detecting the small flow-induced perturbation of the applied magnetic field, which is in the order of several hundred nT, in particular as it is sensitive to ferromagnetic parts close to the sensor and the environmental magnetic field, e.g., generated by currents in cables. This sensitivity can be reduced by developing specialized and highly sensitive coils that measure the spatial gradient of the magnetic field, almost eliminating the influence of magnetic fields that are uniform along the sensor axis [29,30], i.e., Earth’s magnetic field. It has been shown that CIFT can reconstruct the velocity in the presence of a static magnetic field of the EMBr [31,32], and even if the EMBr strength is changed during the measurement [33]. A real-time reconstruction algorithm has been proposed enabling a real-time monitoring of the flow [34].

In this paper, we demonstrate, for the first time, an automatic control of a ruler EMBr based on the CIFT reconstruction of the flow in the mould of the small-scale Liquid Metal Model for Continuous Casting (mini-LIMMCAST) facility at the Helmholtz-Zentrum Dresden - Rossendorf. The model is operated with a eutectic alloy GaInSn, which is liquid at room temperature. The mould has a cross-section of 300×35 mm${}^{2}$ [15,18]. The model is used for the study of flow phenomena in the mould and is operated in isothermal mode. Recently, a systematic parameter study was conducted to investigate the influence of the position of the EMBr on the flow using ultrasound Doppler velocimetry (UDV) [15]. Based on these UDV measurements, we investigated the possibility to develop control loops for mini-LIMMCAST [35].

In order to show the feasibility of a control loop based on CIFT measurements, we equipped the mould of mini-LIMMCAST with a CIFT sensor, which can compensate for the changes of the EMBr strength, and developed a clogging model, which is used to introduce a disturbance in the form of the flow asymmetry. It turned out that this asymmetry is enhanced when the EMBr is active. Therefore, we decided to develop a disturbance rejection controller, which detects the actual impingement position of the jet at both narrow faces of the mould and turns off the EMBr if the difference between the impingement positions is above a critical threshold for a given time.

After a short description of the laboratory model and the operating principles of CIFT, we will explain the compensation of the effects of the EMBr on the CIFT measurement and the influence of the clogging model on the flow in the mould. Then, we describe the design of the controller and the results of the automatic control. Finally, we give an outlook of proposed future work.

## 2. Experimental Setup

### 2.1. Contactless Inductive Flow Tomography

Contactless inductive flow tomography is a method for reconstructing the three-dimensional velocity structure of a conductive fluid by measuring the perturbations of an applied magnetic field caused by the flow structure. Consider the fluid with conductivity σ flowing in a magnetic field B, according to Ohm’s law, an eddy current will be induced:(1)$$j=\sigma \left(B\times v-\nabla \varphi \right),$$
where φ is the potential along the fluid boundaries. The current j gives rise to the secondary magnetic field according to Biot–Savart’s law:(2)$$b\left(r\right)=\frac{\mu_{0}}{4\pi }{\;\int \int \int \;}_{V}j\times \frac{r-r^{\prime }}{{\left|r-r^{\prime }\right|}^{3}}dV,$$
where *V* is the fluid domain, r is a position in space outside the fluid domain and $r^{\prime }$ is a position within the fluid domain.

The divergence free condition for the current density:(3)∇·j=0
must be satisfied. By applying divergence free condition of the current to the Equation (Equation 1), a Poisson equation for the electric potential arises:(4)$$\nabla^{2}\varphi =\nabla \left(v\times B\right).$$

Next, by inserting Equation (Equation 1) into Equation (Equation 2) and resolving the Poisson equation for the electric potential, the following system of equations is obtained:(5)$$b\left(r\right)=\frac{\mu_{0}\sigma }{4\pi }{\;\int \int \int \;}_{V}\frac{\left(v\left(r{\;}^{\prime }\right)\times B\left(r{\;}^{\prime }\right)\right)\times \left(r-r{\;}^{\prime }\right)}{|r-r{\;}^{\prime }{|}^{3}}dV^{\prime }-\frac{\mu_{0}\sigma }{4\pi }{∯}_{S}\frac{\varphi \left(r{\;}^{\prime }\right)n\left(r{\;}^{\prime }\right)\times \left(r-r{\;}^{\prime }\right)}{|r-r{\;}^{\prime }{|}^{3}}dS^{\prime }\;,$$
(6)$$p\left(r\right)\varphi \left(r\right)=\frac{1}{4\pi }{\;\int \int \int \;}_{V}\frac{\left(v\left(r{\;}^{\prime }\right)\times B\left(r{\;}^{\prime }\right)\right)\cdot \left(r-r{\;}^{\prime }\right)}{|r-r{\;}^{\prime }{|}^{3}}dV^{\prime }-\frac{1}{4\pi }{∯}_{S}\frac{\varphi \left(r{\;}^{\prime }\right)n\left(r{\;}^{\prime }\right)\cdot \left(r-r{\;}^{\prime }\right)}{|r-r{\;}^{\prime }{|}^{3}}dS^{\prime }\;,$$
where p(r) is a factor between 0≤p(r)≤1 that is determined by the shape of the boundary and depends on the solid angle of the surface at the position r. $dS^{\prime }$ and $dV^{\prime }$ are surface and volume elements, respectively. In principle, B(r) is the sum of the applied (excitation) magnetic field $B_{0}\left(r\right)$ and the flow-induced magnetic field b(r). Note, however, that the ratio of b(r) to $B_{0}\left(r\right)$ is governed by the magnetic Reynolds number $R_{m}=vl\mu_{0}\sigma$. For our experiment, the characteristic velocity *v* is the inlet velocity of 1.4 m/s and the typical length scale *l* is the diameter of the jet of 15 mm. The electrical conductivity σ of GaInSn is 3.29 MS/m resulting in $R_{m}=0.086$, which is much smaller than 1. Therefore, the problem can be considered linear for our model and the total B in Equations (Equation 5) and (Equation 6) can be replaced by the excitation magnetic field $B_{0}$.

The (linearized) inverse problem is resolved by discretizing the domain and applying the shape functions to the individual elements. Thus, the relationship between the flow-induced magnetic field and the velocity of the fluid can be written in a matrix form:(7)$$b=M\;\tilde{v},$$
where M is a system matrix as derived in [26].

To resolve the linear inverse problem and to reconstruct the velocity field, the following expression must be minimized:(8)$$\min_{\tilde{v}}\left({∥M\;\tilde{v}-\tilde{b}∥}_{2}^{2}+{∥E\;\tilde{v}-v_{\mathrm{inlet}}∥}_{2}^{2}+{∥G\;\tilde{v}∥}_{2}^{2}+\lambda {∥D\;\tilde{v}∥}_{2}^{2}\right)$$

In this functional, the matrix E is a selection matrix for the nodes of the inlet boundary and $v_{\mathrm{inlet}}$ is the velocity at these nodes. G is a matrix that calculates the divergence of the velocity field. The regularization parameter λ, which allows finding a good compromise between minimization of the residuum and the kinetic energy of the flow field, is selected via the L-curve method. Real-time reconstruction is done by precomputing the inverse of the linear equation system resulting from Equation (Equation 8) for the predefined regularization parameter [34] and multiplying it with the vector containing the flow-induced magnetic field measurements during the experiment.

The size of the matrices is determined by the domain discretization and is limited by the computer memory. However, it only mildly affects the accuracy of the reconstruction which is mostly determined by the number of available sensors. Still, even with an infinitely dense sensor coverage of the surface the intrinsic nonuniqueness of the inverse problem, which basically concerns the depth dependence of the velocity distribution [22], can only be mitigated by regularization. Thus, the spatial resolution for quantifying the quality cannot be easily defined since reconstruction quality only partially depends on the grid size. Previous work shows that the structure of the dominating flow field is reconstructed with a reasonable quality [27]. Here, the flow-induced magnetic field was calculated from the numerically determined velocity field and fed to the inverse problem solver for CIFT. The reconstructed velocity field was then compared with the original velocity field. The mean correlation and the mean error was about 75% and 30% for a sensor arrangement of 8 sensors along each narrow faces of the mould (Figure 1 in [27]).

### 2.2. Mini-LIMMCAST

Experiments were performed at the mini-LIMMCAST facility at Helmholtz-Zentrum Dresden-Rossendorf. Mini-LIMMCAST is a 1:5 scaled isothermal model of a continuous caster and is shown in Figure 1. It is operated with the eutectic alloy of gallium–indium–tin (GaInSn), which is liquid at room temperature. The liquid metal is stored in the catchment tank, from where it is pumped to the tundish using an electromagnetic pump. The level of the liquid metal in the tundish is continuously kept constant by controlling the speed of the pump. From tundish, the metal flows through the SEN into the mould, and the flow rate is controlled by the position of the stopper rod. The mould is made out of acrylic glass and has a rectangular profile of 300×35 mm${}^{2}$ with a height of 620 mm. The SEN has an inner diameter of 12 mm, an outer diameter of 21 mm and two side ports directed downward at an angle of 15${}^{\circ }$. The high electrical conductivity of GaInSn is comparable to the one of liquid steel and enables the use of electromagnetic actuators to alter the flow structure in the mould. Figure 1 shows the sketch of the experimental setup with relative positions of SEN, mould, EMBr of ruler-type, and CIFT measurement and excitation coils.

The EMBr influences the flow by generating a strong magnetic field which induces eddy currents in a flowing metal. Because of the mutual interaction of the magnetic field of the EMBr and the induced currents, the liquid metal will experience Lorenz forces opposite to the direction of the fluid flow, essentially braking the fluid and altering the flow pattern. EMBr is powered by a maximum current of 600 A, generating a magnetic flux density up to 404 mT.

Measurements of the velocity at the laboratory setup can be performed either with CIFT or UDV. While UDV provides linear profiles of the velocity component in the direction of the ultrasound beam, it is a reliable and well-established measurement method for liquid metals. By arranging multiple ultrasound transducers in an array, a scalar field can be constructed containing just one velocity component. CIFT can reconstruct the full two-dimensional velocity field. For this purpose, two excitation coils are installed, one above and one below the ferromagnetic yoke of the EMBr. To measure the flow-induced magnetic field, fourteen gradiometric coils were placed, seven on each narrow side of the mould. The gradiometric coils consist of two counterwound coils connected in series. This configuration is robust in discarding the effects of uniform magnetic fields on the measurement, e.g., Earth’s magnetic field.

The excitation magnetic field is generated by an alternating current (AC) with frequency of 8 Hz. By using AC excitation magnetic field, the amplitude of the flow-induced magnetic field is encoded at the same frequency and in-phase to the excitation magnetic field. The demodulation is done by applying the Lomb–Scargle algorithm on the measured voltage [36]. The flow-induced magnetic field is calculated from the sensor transfer function and the measured induced voltage.

For the sake of simplicity, only one regularisation parameter (λ in Equation (Equation 8)) is selected and used for all real-time reconstruction, because it turns out that the values of the regularisation parameter for a variety of experiments are nearly constant. From the reconstructed velocity field, the information of the jet impingement position is determined by finding the area within 10 mm from the narrow wall, where the average vertical velocity is zero ${\bar{v}}_{z}$ = 0 mm/s.

## 3. Results

### 3.1. Effects of the EMBr on CIFT

When it comes to implementing CIFT as feedback in the control loop, several challenges arise. First, the compensation of the influence of the EMBr on the measurements is not straightforward, as it needs to compensate for the hysteresis exhibited by the ferromagnetic parts of the EMBr. The effect of the EMBr is shown in Figure 2. Figure 2a shows the measured flow-induced magnetic field for an experiment where the EMBr current was changed during the run. It can be seen that the flow-induced magnetic field changes drastically when the current through the EMBr is varied from 0A to 200A at t≈290 s. This phenomenon can be explained by the ferromagnetic properties of the yoke, which depend on the magnetic field generated by the coils of the brake. Therefore, the shape of the excitation magnetic field, which is partly closing through the yoke, also changes. This alteration is visible as a static offset of the flow-induced magnetic field. Figure 2b shows the flow induced magnetic field after the compensation procedure described in the following.

One of the properties of the magnetic hysteresis is the congruency property. This property states that all minor hysteresis loops corresponding to the same extreme input values are congruent in the geometrical sense [37]. So it can be expected that for two consecutive experiments, for the same change of the EMBr current, the change of the measurement of the flow-induced magnetic field would also be the same. Utilizing this property is easy, but it is required to perform identification measurements before the measurements with fluid flow can be started.

Complex control strategies often require the actuator to assume any state, limited only by the minimum and maximum values. However, during the transitions from one set-point to another, comparably high voltages are induced in the measurement coils, and information about the flow-induced magnetic field is lost. Thus, we limit the set-point values for the EMBr current $I_{EMBr}$ to discrete levels with steps of 25 A, $\Delta I_{EMBr}$ = 25 A. Another point to consider when deciding on the congruence-based compensation is that it can only be used reliably for identified values and the transition stages. To cover the entire operation span of the EMBr current, identification should be performed by cycling the EMBr current from minimum to maximum, and back to minimum. Subsequently, only the same current values, and transitions can be used for control. Consider the initial state when EMBr is switched off, and the controller sets the first set-point of the EMBr current higher than zero. If the second set-point is larger than the first one, the EMBr current must first increase to the maximum and then decrease to its new set-point. This mode of operation could introduce some undesirable effects on the flow, and it is not well-suited for complex control strategies.

A more flexible compensation method is by implementing the numerical model of the hysteresis. For this purpose, the Krasnosel’skii–Pokrovskii (KP) [38] model of hysteresis was implemented that performs the compensation in real-time. However, due to the lack of precise EMBr current source and measurement method, the error introduced from the compensation, in some cases, is larger than, or in the same range of the expected flow-induced magnetic field. Figure 3 shows the compensation when using a hysteresis model for a given identification and measurement current steps. During identification, the corresponding mean value of the flow-induced magnetic field was recorded for all sensors for every current step. The measurements were used to identify the weights of the KP model using a discrete model and least-squares method adopted from Stakvik et al. [39]. With the determined weights of the system, the prediction of compensation can be calculated from a given set of input currents. However, even though the absolute error for the sensor shown is in order of 20 nT, the expected value of the flow-induced magnetic field is in the same order of magnitude. The underlying uncertainty makes it difficult to reconstruct the velocity field reliably, and even though the model can be improved, for the first tests of the real-time control a congruency-based compensation was used for just one value of the EMBr current, limiting the control strategy to an on/off controller. The implemented numerical KP hysteresis model inherently supports this operating mode, and with further improvements, it can be used for more complex controllers.

### 3.2. Control Strategy

Previously developed control strategies that are based on UDV measurements [35] could not be directly transferred to CIFT-based control because of the different spatial and temporal resolution of both measurement modalities. UDV provides a finer temporal and spatial resolution along the measurement line, and it is not influenced by the changes of the magnetic field strength of the EMBr. However, the measurements were only conducted on one side of the mould. On the other hand, CIFT can provide a two-dimensional velocity field for the entire fluid domain, albeit with relatively sparse resolution which is basically related to the limited number of magnetic field sensors and the intrinsic non-uniqueness of CIFT. The amount of nodes at which velocities are reconstructed is limited by memory of the computer, and for our experiments it was in the order of 10,000 nodes, and the grid spatial resolution in the upper mold area is the range between 5 and 10 mm.

In this study, an obstacle was introduced into the flow to generate a disturbance in form of an asymmetric flow that can be detected by CIFT. The drawing of the obstacle and its position with respect to the SEN is shown in Figure 4. As in [40], the obstacle serves to simulate the situation of SEN clogging, which can be realized by reducing the port size resulting in lower velocities at the clogged side and increased velocity and deeper impingement on the opposite side. The obstacle used has a bevel with a 45° angle to ensure the asymmetry of the jet impingement position is significant enough for detection during the real-time reconstruction. The influence of the obstacle on the reconstructed flow field can be seen in Figure 5. Figure 5a presents the reference flow without inserted obstacle and without active EMBr. Figure 5b is the reconstruction of the experiment when the current through EMBr is set to $I_{EMBr}$ = 200 A; for this range of operation of the EMBr, there is no significant effect on the jet impingement point [15]. However, if an obstacle is placed at the SEN outlet at a certain position, the flow rate on one side is reduced, and the jet impingement point is deeper in the mould, visible from Figure 5c. If the EMBr is then switched on, the jet seems to impinge deeper in the mould, enhancing the asymmetry, as shown in Figure 5d. This increased asymmetry is the scenario which the controller is trying to identify and respond to.

Due to the limitations on compensating the effect of the EMBr as described in the previous section, we constrain the operation of the EMBr to two modes of operation, on-state with $I_{EMBr}$ = 200 A and off-state. This constraint significantly reduces the training time, from several hours for a training sequence from Figure 3a, to a few minutes for the simpler two state changes from 0 A to 200 A and back to 0 A. The selected scenario is realistic since, in the industrial process of steel casting, the operating current of the EMBr is predetermined by the product recipe and is empirically selected, either from plant measurements or numerical simulations. However, in a steel caster, the SEN’s clogging generally increases over time, but for simplicity, we model it as an invariant occurrence.

The reconstructions in Figure 5 show instantaneous velocity profiles in the center cross-section of the mould for four instants of time. The reconstructed velocity fields are obtained by solving the minimization problem in Equation (Equation 8). Due to the turbulent nature of the jet, the impingement position has to be time-averaged. Figure 6 shows the time average of the jet impingement points on both narrow sides of the mould. It can be seen that the mean jet impingement point on both sides do not differ when there is no clogging, even when the EMBr is turned on. When clogging is introduced, a difference of about 5 mm can be observed in the time interval between 300 and 500 s. This difference increases significantly to 20 mm, when the EMBr current is set to $I_{EMBr}$ = 200 A. It is clear that this condition is unwanted during casting. Based on the observed interaction between EMBr, obstacle, and the flow, an on/off controller is designed that detects the strong asymmetry in the jet impingement position, classifies it as a result of clogging, and switches off the EMBr.

The design of the controller can be seen in Figure 7. It receives information about the jet impingement at both narrow faces of the mould from the real-time reconstruction every 0.5 s. A running mean filter is then applied, and the difference between the impingement positions of the jet on both sides is calculated. The absolute difference between the two is an input to the *Relay operator* whose output is set to logical 1 if the difference is larger than the value of the *Impingement Diff On* input variable, in this case, more than 20 mm. The output is reset if the absolute difference is lower than the value of *Impingement Diff Off* input variable. If the output from the relay operator is 1 for a set amount of time, given by the *Time Delay*, it is evaluated with two additional conditions: the flow must be active, set by the operator with the toggle switch, and current must equal the desired value. The two additional conditions are necessary only for ensuring stability and control over the experiment. If all the conditions are satisfied, a state is set that automatically changes the current set-point to the EMBr to the desired value given by constant *Control action*, in this case, to 0 A. A controller reset has to be performed by the operator to avoid any toggling of the current set-point by setting the *Manual Current Setpoint* to the same value of *Control action*. The current set-point is sent to the current source for the EMBr. The complete communication diagram and block scheme of the controller is given in the Appendix A.

In order to prove the effectiveness of the control loop, a similar experimental run was started with an active control loop. The goal was to show that the controller is able to automatically detect and react to the asymmetry as a result of interaction of EMBr and the clogging model. The controller changes the set-point of EMBr current to the predetermined value for which the hysteresis model was trained. The recording of this experiment is shown in Figure 8. Once the experiment has started and the flow is developed, the EMBr current is set to $I_{EMBr}$ = 200 A at t≈500 s. A significant asymmetry cannot be observed. After some time, we introduce the obstacle to the SEN outlet. The obstacle does not have a fixed position, but rather, it is lowered until the asymmetry occurs. At t≈620 s a strong asymmetry becomes apparent. If the asymmetry condition is satisfied for 15 s, the controller changes the set-point value to $I_{EMBr}$ = 0 A at t≈700 s. A significant reduction in asymmetry is noticeable, and we conclude that the controller successfully executed the desired action. We repeated the test case starting at t≈900 s during the same measurement in order to validate the repeatability.

## 4. Conclusions and Outlook

The lab demonstration for the continuous casting model showed for the first time that an electromagnetic actuator can be controlled based on the actual flow structure in the mould. It is shown that CIFT can be used to monitor the flow in real time even if the EMBr changes its strength, and it can be integrated into a control loop for an EMBr. The major challenge remains the accurate compensation of the effects of the change of EMBr current on CIFT. These effects are up to three orders of magnitude larger than the flow-induced signal.

As a first test case, nozzle clogging was simulated by introducing an obstacle and partially blocking one of the outlet ports of the SEN, thus deflecting the flow. Under these circumstances, the impingement position of the jet moves downward when the EMBr is active. This undesired change of the jet impingement position can be corrected by switching off the EMBr. With this proof of concept, a working control loop implementation is available form now on, which is a valuable tool to investigate more sophisticated control strategies.

CIFT offers unique insights into the flow structure of the mould, and with that, a new way to monitor and control the process. The current controller implementation, albeit simple in nature, shows significant progress in using tomographic measurement techniques as a controller backbone. Further improvement of control strategies consists of increasing the measurement accuracy, robustness and speed, and identifying additional key flow features of interest for an efficient process. The next logical step is to implement a proportional–integral–derivative (PID) controller and gradually increasing the complexity of the control algorithm, as one of the popular methods for tuning PID controllers is based on the relay control similar to what is shown in our experiments [41]. The PID controller could be used for controlling the impingement point of the jet in order to keep it in the certain range. This method of control could be then easily compared with the UDV measurements to quantify the accuracy and quality of compensation, reconstruction, and control.

Utilizing more complex control strategies will primarily require improving the compensation of the effects generated by the ferromagnetic parts. The two-way coupling of the reconstruction and compensation by the controller can result in unstable states and further precautions must be made to avoid them. This can be achieved by further enhancing the magnetization model or, perhaps, developing an additional inductive sensor, which detects the state of the magnetization vector in the yoke.

Future investigations should focus on generating a map of typical flow instabilities, which might be induced by higher liquid metal flow rates, Argon gas flow rates, nozzle clogging, bulging of the strand, etc. New control strategies can be achieved by using more sophisticated actuators, like local EMBrs, which independently influence the flow on both sides of the mould, or electromagnetic stirrers at the free surface. Such electromagnetic actuators are already available for continuous casters. Furthermore, new arrangements of the excitation coil and the magnetic field sensors for CIFT should be investigated. Last but not least, a more sophisticated solution of the inverse problem is pursued.

Further improvements might become possible by combining different tomography methods, so that key flow features could be mapped to flow instabilities and a controller could be upgraded. An example of this is the classification and identification of two-phase flows in the SEN using MIT. However, MIT system requires to have high frame rate to capture the flow dynamic in the SEN. In any case, the main advantage of CIFT and MIT is that, because of their contactless nature, they can be implemented in aggressive industrial environment of steel casting, and provide information that was previously unavailable in order to improve the process control and design.

## Figures and Tables

**Figure 1 sensors-22-02195-f001:**
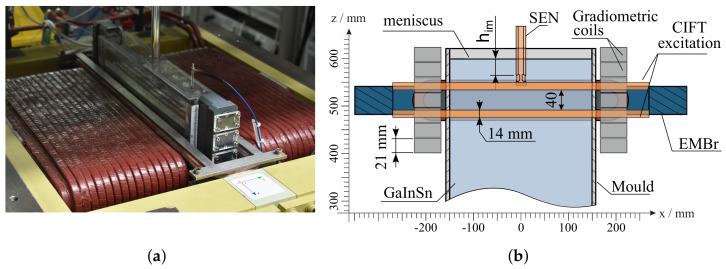
Experimental setup containing contactless inductive flow tomography (CIFT) coils and sensors and the electromagnetic brake (EMBr). Two excitation coils generate a primarily vertical magnetic field. Fourteen gradiometric coils, seven on each narrow side, are used to measure the flow-induced magnetic field. EMBr generates a strong magnetic field below the submerged entry nozzle (SEN), perpendicular to the wide side of the mould. (**a**) Photograph of the mould and CIFT coils. (**b**) Sketch adopted from Schurmann et al. [15].

**Figure 2 sensors-22-02195-f002:**
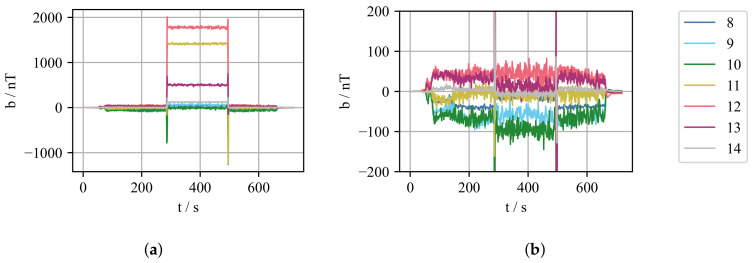
Measurement of flow-induced magnetic field during the experiment when the EMBr current changed. (**a**) Uncompensated flow-induced magnetic field, during the experiment the EMBr current changed from 0 A to 200 A at t≈290 s and back to 0 A at t≈500 s. (**b**) Compensated flow-induced magnetic field.

**Figure 3 sensors-22-02195-f003:**
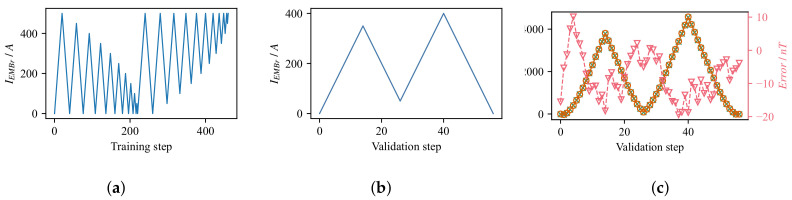
Compensation error of the Krasnosel’skii–Pokrovskii (KP) hysteresis model. (**a**) Current profile used for identification of the model weights; every measurement step represents a period of fifteen seconds, during which the mean value of the flow-induced magnetic was recorded for each sensor as an input to the hysteresis model. Current is changed in steps of $\Delta I_{EMBr}$ = 25 A. (**b**). Current profile used to validate the accuracy of the model. For each step the corresponding value of the flow induced magnetic field is recorded and compared to the output of the model in case of the use of same current profile as an input to the model. (**c**) Absolute error of the model shown for the middle gradiometric coil on the right side of the mould (Sensor 4).

**Figure 4 sensors-22-02195-f004:**
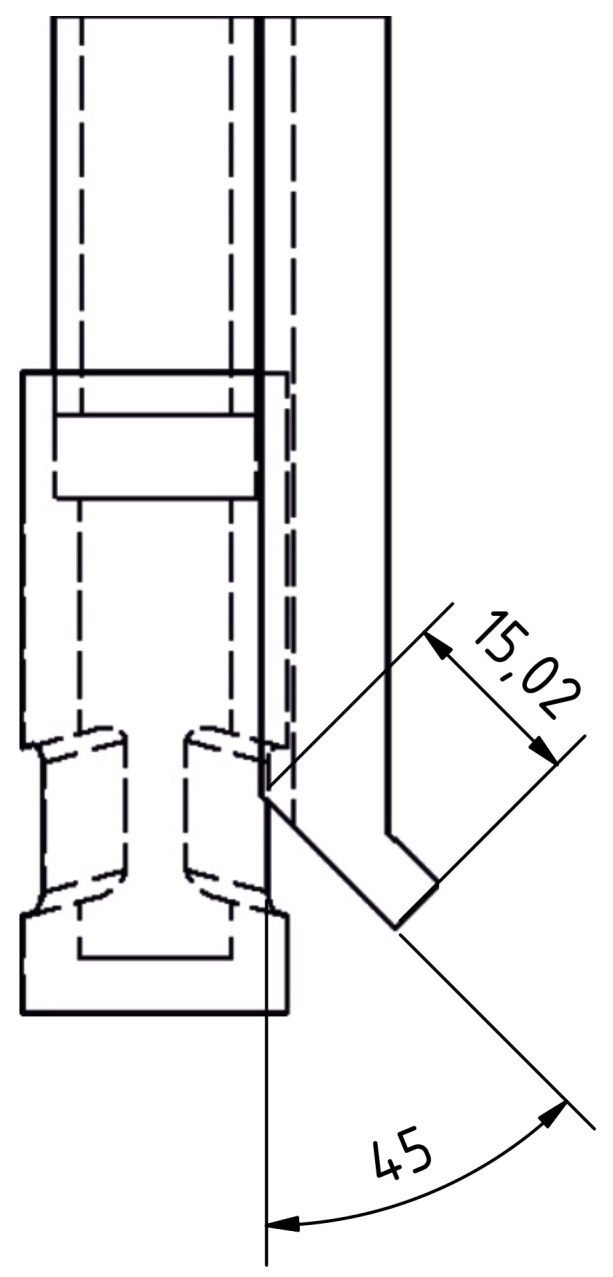
Setup of the clogging model showing the obstacle position with respect to the SEN. The obstacle is positioned to cover half of the SEN outlet and has an additional 15 mm long extension directed downward at an angle of 45°. Outlets of the SEN are directed downward at an angle of 15°.

**Figure 5 sensors-22-02195-f005:**
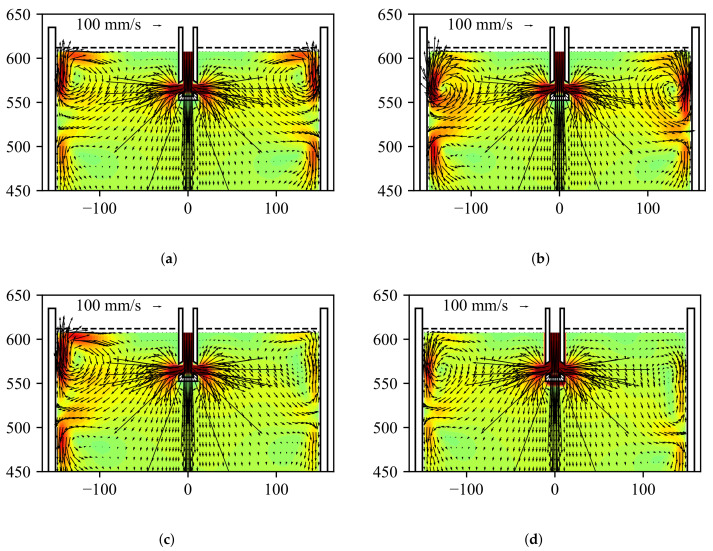
Reconstructed velocity fields in the center xz-plane of the mould for different experimental configurations. (**a**) No clogging; $I_{EMBr}$ =0 A. (**b**) No clogging; $I_{EMBr}$ = 200 A. (**c**) With clogging; $I_{EMBr}$ = 0 A. (**d**) With clogging; $I_{EMBr}$ = 200 A.

**Figure 6 sensors-22-02195-f006:**
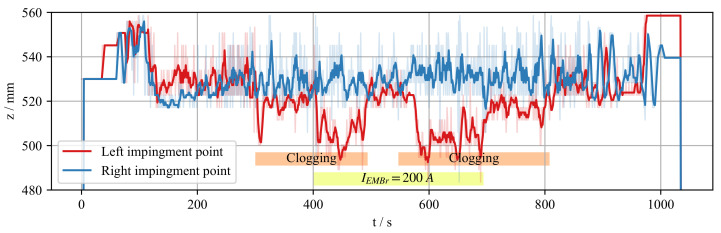
Jet impingement position at both sides of the mould. Less opaque background graph shows the raw position of the impingement point. Solid lines show the running average of the respective impingement point. Running mean window is ten samples (5 s).

**Figure 7 sensors-22-02195-f007:**
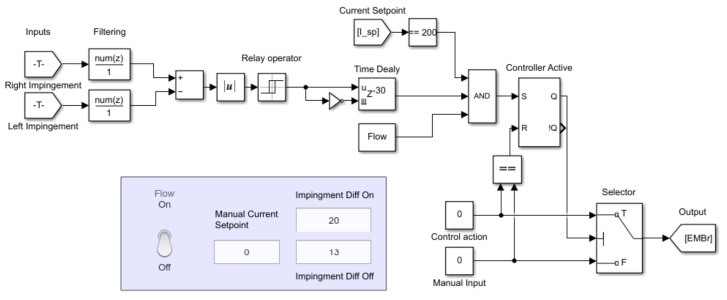
Real-Time SIMULINK implementation of the controller. The controller evaluates the difference of the jet impingement points on both sides. If the difference is within a window defined by the relay operator for 15 s, the controller will set a new EMBr current if the experiment is in operation, and EMBr current is at a valid set-point. Controller is reset and ready for new evaluation when the operator set a manual current set-point to be equal the controller value.

**Figure 8 sensors-22-02195-f008:**
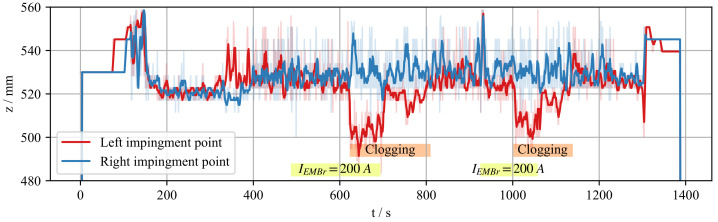
Controller validation experiment. The flow reaches a stable state by t=200 s. The same scenario is repeated twice during the same experiment. First, the EMBr current is set to $I_{EMBr}$ = 200 A at t≈500 s and t≈900 s.

## Data Availability

Data supporting the reported results is available as open access doi:10.14278/rodare.1463 (accessed on 2 February 2022).

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
