# Peer review of "Laboratory Investigation of Tomography-Controlled Continuous Steel Casting"

_sensors, 2022, doi:10.3390/s22062195_

Round 1

Reviewer 1 Report

This work is meaningful and interesting, but the following revisions are needed:

  1. This title “Tomography-controlled continuous steel casting “is a bit big. More specific title is needed.
  2. Could you give some two dimensional flow tomography results based on CIFT?
  3. How to get the results from Figure 5?
  4. Beside clogging of the Submerged Entry Nozzle, is there any other method to validate the new method in this paper?
  5. The temporal resolution of CIFT is 1 second. How about the spatial resolution?

Author Response

Dear reviewer, Thank you very much for your time and effort. See attached corrected manuscript.

This title “Tomography-controlled continuous steel casting“ is a bit big. More specific title is needed.

The aim of work was to introduce the new control concept which is not necessarily restricted to laboratory environment. However, you are right, and we understand the title is misleading. Subsequently, the title of the publication has been altered to better reflect its contents. "Laboratory investigation of tomography-controlled continuous steel casting"  

Could you give some two dimensional flow tomography results based on CIFT? How to get the results from Figure 5?

Figure 5. shows a few frames of the 2D reconstructed velocity fields obtained by Contactless Inductive Flow Tomography. The rest of reconstructions from which we extracted the jet impingement position for Figure 6 and Figure 8 are (soon) available in as data files and images. Additionally, we  referenced in the description of the figure 5  that it is a result of minimizing equation (8) and resolving for velocity v. In order to make this clearer in the text, we added the following sentance: "The reconstructed velocity fields are obtained by solving the minimization problem in Equation (8)"  

Beside clogging of the Submerged Entry Nozzle, is there any other method to validate the new method in this paper?

The aim of the work was to test and demonstrate that CIFT has the potential to act as a sensing device for a control loop. The elementary challenge of this approach is the robust and reliable compensation of the effects of changing the strength of the electromagnetic actuator on the magnetic field measurements for CIFT. For a first proof of principle we selected a simple, but yet realistic example modeling nozzle clogging. The effect is easily repeatable and has a significant impact on the flow. Therefore, it facilitates the development of a control loop. Of course, nozzle clogging is not the only scenario, where CIFT can be applied. In the curse of the investigations, we also tried to introduce significant disturbances in the flow by enlarging the throughput of liquid metal and injecting Argon gas in the Submerged Entry Nozzle. However, in the investigated parameter range the flow remained rather stable so that a control action was not required. Of course, a more sophisticated and detailed investigation is needed for the analysis of flow regime changes and for the development of compensation strategies with an EMBr. We believe that the method should be also applicable for more complex scenarios, including different kind of actuators, like double-ruler and local brakes. We are currently working on expanding the range of operation of EMBr to control the position of the jet regardless of the clogging which can be validated with UDV technique. We updated the discussion section with this information  

The temporal resolution of CIFT is 1 second. How about the spatial resolution?

The exact specification of the spatial resolution for CIFT is a challenging task, because of the non-uniqueness of the underlying inverse problem. The structure of the dominating flow field is reconstructed with a reasonable quality as discussed in reference 28. In a numerical study the flow induced magnetic field was calculated based on a CFD determined velocity field and fed to the inverse problem solver for CIFT. The reconstructed velocity field was then compared with the original velocity field. The mean correlation and the mean error was about 75\% and 30\% for a sensor arrangement of 8 sensors along each narrow faces of the mould (see Figure 8 in reference 28). Furthermore, we compared also the mean velocity profile at different heights close to the jet in the mould between CIFT and UDV (see figure 17 and 18 in reference 28). Therefore, an exact value cannot be given which is true for the entire fluid volume. A more sophisticated analysis of the accuracy of the reconstruction is needed, which is beyond the scope of the present paper and will be addressed in the future.    

Reviewer 2 Report

Dear Authors I congratulation, excellent work. However some additional data should be add to paper.

  • The quantities for magnetic Reynolds number should be explained in the text of paper.
  • To sketch 1 should be add coordinate system due to average vertical velocity.
  • Description of Y axis in the figure 3b should be add.
  • Above 40% of cited literatures are the works of Authors work, therefore some additional references on continuous steel casting process should be add to paper.

Author Response

Dear reviewer, Thank you very much for your time and effort. See attached corrected manuscript.

The quantities for magnetic Reynolds number should be explained in the text of paper.

We updated the section with the values of conductivity, velocity and length scale and did rough estimate of the Rm for our experimental setup. "For our experiment, the characteristic velocity $v$ is the inlet velocity of 1.4 m/s and the typical length scale $l$ is the diameter of the jet of 15 mm. The electrical conductivity $\sigma$ of GaInSn is 3.29 MS/m resulting in $R_m = 0.086$, which is much smaller than 1."  

To sketch 1 should be add coordinate system due to average vertical velocity.

Another reviewer commented on the figure, and we updated it with coordinate system and the most important dimensions.  

Description of Y axis in the figure 3b should be add.

Figure has been updated. Thank you.  

Above 40\% of cited literatures are the works of Authors work, therefore some additional references on continuous steel casting process should be add to paper.

We have added additional references regarding the continuous casting, and removed some redundant references of the authors.

Reviewer 3 Report

A very interesting approach to control the process of continuous casting, even though this is done on a lab-scale and the proposed controller is quite simple in its design (on/off). However, the latter point is justified considering the physical challenges the sensor (CIFT) and actuator (EMBr) face during joint operation. I suggest to accept the paper for publication after the following points are addressed:

  • figure 2: what is the legend scale? height in the mould / coils of the CIFT?
  • figure 6 and explanations (lines 346-359). I am not sure that all explanations refer to figure 8. For instance, "the controller changes the set-point value to IEMBr = 0 A at t 500 s". Isn't it t 700 s when we look at figure 8? You also mention that the test case is repeated at t 700 s. Isn't it t=1000 s instead? Please double-check your explanations for consistency with the figure.
  • Appendix A (figure A2): what is LTT24? Also, it is mentioned about MIT reconstruction. However, MIT is not used in this paper, just mentioned as a possible solution for flow identification in the SEN. Please consider either extending the text or modifying the figure.

Also some (very) minor changes:

  • line 246: valuees (delete one "e")
  • line 402: "MIT system requires has to have high frame rate402
    to capture the flow dynamic in the SEN". Pleae correct the english.
  •  

Author Response

Dear reviewer, Thank you very much for your time and effort. See attached corrected manuscript.   figure 2: what is the legend scale? height in the mould / coils of the CIFT? Figure 2 was updated with the dimensions and coordinate system  

figure 6 and explanations (lines 346-359). I am not sure that all explanations refer to figure 8. For instance, "the controller changes the set-point value to IEMBr = 0 at t  500 s". Isn't it t 700 s when we look at figure 8? You also mention that the test case is repeated at t 700 s. Isn't it t=1000 s instead? Please double-check your explanations for consistency with the figure.
The values in the manuscript were wrong, and it was corrected. Thank you.

Appendix A (figure A2): what is LTT24? Also, it is mentioned about MIT reconstruction. However, MIT is not used in this paper, just mentioned as a possible solution for flow identification in the SEN. Please consider either extending the text or modifying the figure. LTT24 is the A/D converter that we use to measure the voltage induced in the coils. We added this information in the appendix. MIT reconstruction was not used in the presented experiments and the reference was removed from the figure.   Also some (very) minor changes:
line 246: valuees (delete one "e") line 402: "MIT system requires has to have high frame rate402 to capture the flow dynamic in the SEN". Please correct the english.} Thank you. It has been corrected.  
